# A study protocol for a multi-specialty observational cohort comparing robotic stapler and bedside stapler outcomes in robotic-assisted surgeries

**Yu-Ting Chi[1], Naomi C. Hamm[1], Shih-Hao Lee[1], Minkyung Shin[1], Yuki Liu[1], I-Fan Shih[1], Feibi Zheng[1,2], Ben Forrest[1], Peng-Lin Lin**[1]*

**1** Health Economics & Outcomes Research, Intuitive Surgical, Sunnyvale, California, United States of America, **2** DeBakey Department of Surgery, Baylor College of Medicine, Houston, Texas, United States of America

* penglin.lin@intusurg.com

## Abstract

### Introduction

Surgical staplers are essential tools in minimally invasive surgery (MIS), enabling tissue division, hemostasis, and secure anastomoses. With the growth of robotic-assisted surgery, robotic staplers such as SureForm have recently become available. These staplers offer precise articulation and real-time tissue compression monitoring. However, the clinical advantages of robotic staplers over bedside staplers remain uncertain. Studies show mixed results across specialties, mainly due to small sample sizes, outdated data, and data heterogeneity. This study protocol proposes a series of future analyses that will evaluate the clinical outcomes and resource utilization of robotic versus bedside staplers in robotic-assisted surgeries across multiple special-ties using recent real-world data.

### Methods and analysis

This retrospective cohort study will use data from the Premier Healthcare Database (PHD), a large hospital-based database covering patients with varied payers across the United States. Adult patients (≥18 years) who underwent elective, fully robotic-assisted lung, colorectal, gastric, or bariatric surgeries from 2019 to 2023 will be included. Each surgical specialty will be analyzed in a separate paper. Patients will be categorized into two groups based on the type of surgical stapler used: robotic staplers (SureForm) and bedside staplers (manual or powered). The primary outcome will be postoperative leak (air leak for lung resection; anastomotic leak for colorectal, gastrectomy, and bariatric). Key secondary outcomes are other complications, conversion to open surgery, operative time, transfusion requirements, length of stay (LOS), and cost. Overlap weighting will be applied to minimize bias.

which permits unrestricted use, distribution, and reproduction in any medium, provided the original author and source are credited.

**Data availability statement:** The data used in this study are from the Premier Healthcare Database (PHD), a commercial dataset owned by Premier, Inc. The authors do not have permission to share the raw data. Qualified researchers may request access to the PHD directly from Premier, Inc. (https://premierinc.com/transforming-healthcare/healthcare-data). The authors had no special access privileges. All analytic details needed to reproduce the study are included in the manuscript and supporting information files.

**Funding:** The author(s) received no specific funding for this work.

**Competing interests:** All authors are employees of Intuitive Surgical. Y-T Chi, N C Hamm, S-H Lee, M Shin, Y Liu, I-F Shih, F Zheng, B Forrest, and P-L Lin receive salaries from Intuitive Surgical. Additionally, S-H Lee, M Shin, Y Liu, I-F Shih, F Zheng, B Forrest, and P-L Lin hold restricted stock units (RSUs) from Intuitive Surgical. This does not alter our adherence to PLOS ONE policies on sharing data and materials.

## Dissemination

Results will be disseminated through peer-reviewed surgical journals and presentations at relevant surgical meetings.

## Introduction

Surgical staplers are essential instruments in minimally invasive surgery (MIS), enabling surgeons to effectively perform tissue division, hemostasis, and create secure anastomoses. With the increasing adoption of robotic-assisted surgery, stapling technology has advanced to offer enhanced capabilities. Bedside staplers include manual staplers, which require direct force application by surgeons, and energy powered staplers, which provide more consistent staple formation. Over the past decades, these staplers have evolved to include features such as three-dimensional stapling and automatic force delivery, which reduce operator variability and physical strain while increasing precision [1]. Recently, robotic staplers, such as the SureForm stapler developed by Intuitive Surgical, have introduced advanced features including precise articulation, automated real-time tissue compression monitoring, and the ability for surgeon-controlled firing from the robotic console [2]. While these innovations show potential to improve clinical outcomes and operative efficiency [3], their effectiveness compared to bedside staplers remains unclear.

Recent literature presents mixed findings regarding the clinical benefits of robotic staplers across various surgical specialties. In thoracic surgery, Zervos et al. reported that using robotic staplers during robotic lobectomy was associated with less intraoperative bleeding and lower conversion rates to open procedures compared with bedside staplers [4]. In contrast, Phillips et al. found no significant differences in peri-operative outcomes between robotic and bedside staplers [5]. In colorectal surgery, robotic staplers appear to facilitate technically challenging low rectal transections, potentially lowering rate of symptomatic anastomotic leakage compared to bedside staplers, as well as reducing the number of staple firings [6–9]. However, a 2020 meta-analysis by Tejedor et al. did not find a statistically significant reduction in anastomotic leak rates [10]. Bariatric studies suggest that robotic staplers are associated with increased operative times without clear advantages in preventing staple-line complications [11–13]. For oncologic gastrectomy, most studies suggest that robotic techniques lead to better clinical outcomes, such as reduced bleeding, longer distal resection margins, earlier time to oral intake, and fewer complications, though they are associated with higher costs when compared with laparoscopic techniques [14,15]. However, the impact of different staplers on their own remains unknown in robotic gastrectomy.

Economic evidence on stapler comparison is limited, with only a few studies examining hospital costs associated with robotic and bedside staplers. Notably, there is only one cost-effectiveness study each for lung and colorectal surgeries. Zervos et al. found no significant cost differences between robotic and bedside staplers in robotic lobectomy. In contrast, Holzmacher et al. reported significantly lower per-patient costs for robotic staplers in colorectal surgeries [4,6]. However, most studies in bariatric

surgery indicate that robotic staplers are associated with higher device-related costs [11,13]. These findings highlight the need for further economic evidence to guide healthcare decision-making and resource allocation.

Despite the body of literature discussed above, significant gaps remain across various specialties. First, most studies looking at stapler specific outcomes analyzed data collected prior to 2019, before the commercial launch of Sureform stapler. Consequently, no thoracic and colorectal studies have compared SureForm with laparoscopic bedside staplers; they only evaluate the earlier generation EndoWrist robotic stapler. A prospective study from India is the only one that examines the clinical outcomes of the SureForm SmartFire stapling system in robotic sigmoid colon and rectal procedures, demonstrating its technical performance, such as the 120-degree angulation capability for precise tissue transection, and reporting no leakage or bleeding [16]. Second, the literature shows diverse approaches in stapler comparisons: some studies compare robotic staplers exclusively with manual staplers [5,7–9], others with powered staplers [11], and some with both [4,6,11,13]. Additionally, most studies are limited by small sample sizes. Wide confidence interval in the bariatric study by Clapp et al. also resulted in unstable findings [13]. These limitations highlight a significant gap in the evidence regarding stapler comparisons for robotic-assisted surgery. Furthermore, many existing datasets have high heterogeneity and limited data granularity, which limits the ability to conduct robust stapler comparison studies.

Due to these conflicting findings and the absence of robust multi-specialty evidence, this study protocol outlines a comprehensive, real-world comparative research program to determine whether robotic staplers, specifically SureForm staplers, provide measurable clinical advantages over bedside staplers in robotic-assisted surgeries. Thus, the objective of this study protocol is to prespecify large-scale real-world analyses evaluate perioperative clinical outcomes and healthcare resource utilization associated with robotic and bedside staplers. This protocol prespecifies a series of comparative analyses in robotic-assisted surgeries in multiple specialties such as thoracic, colorectal, gastric, and bariatric surgeries.

## Methods and analysis

### Study design

This protocol describes a retrospective, observational cohort framework that will use data from the Premier Healthcare Database (PHD; formerly known as the Premier Healthcare Database). PHD is a comprehensive U.S. hospital-based, Health Insurance Portability and Accountability Act (HIPAA)-compliant database comprising detailed inpatient and outpatient information from over 1,400 hospitals across diverse geographic and demographic settings [17]. It contains more than 352 million unique patients, with over 1.4 billion outpatient visits and 172 million inpatient visits. Annually, since 2012, it has recorded over 86 million outpatient visits and nearly 9 million inpatient visits, capturing approximately 25% of the annual inpatient admissions in the United States [18]. The protocol specifies inclusion of adult patients (≥18 years) who underwent elective, fully robotic-assisted lung, colorectal, gastric, or bariatric surgical procedures between January 1, 2019, and December 31, 2023.

Each surgical specialty will be analyzed separately and reported in separate publications. The anticipated timeline for data extraction and analysis is as follows: For lung and colorectal resections, data extraction is planned to begin on September 1, 2025, with data collection expected to be completed by October 31, 2025, and results available by December 31, 2025. For bariatric surgeries and gastrectomy, data extraction will commence on January 1, 2026, with completion anticipated by February 28, 2026, and results available by April 30, 2026. As of the time of protocol submission, data extraction and analysis have not yet begun. All data accessed from the Premier Healthcare Database are fully de-identified; the authors will not have access to any information that could identify individual participants during or after data collection.

### Study Population

Patients will be identified from the PHD using the International Classification of Diseases, Tenth Revision, Procedure Coding System (ICD-10-PCS) and Current Procedural Terminology (CPT) codes specific to robotic-assisted lung, colorectal,

gastric, and bariatric procedures. Each surgical specialty will be analyzed separately and reported in separate publications. Inclusion for each specialty-specific cohort will require both a robotic procedure code and a specialty-specific surgical code within the same encounter. Codes that will be used to define cohorts are in **Table 1**.

Only the first robotic-assisted procedure per specialty will be included for each patient to ensure a single primary surgical encounter is analyzed. Only primary surgical procedures will be included in this study. Patients will be excluded if they are missing critical operative data, such as details related to stapler usage documentation, or usage of robotic staplers other than SureForm (**Fig 1**).

Eligible patients are categorized into two distinct exposure groups based on the type of surgical stapler utilized during surgery: (1) Robotic Stapler Group, which consists of patients who underwent surgical procedures using SureForm robotic staplers. This group includes patients who exclusively used robotic staplers as well as those who had a mixed usage of robotic and other staplers; and (2) Bedside Stapler Group, which comprises patients who received surgical procedures without SureForm robotic staplers, relying exclusively on staplers manually operated by a bedside assistant. Detailed information regarding the algorithm used to classify patients can be found in the *Exposure Assessment: Surgical Stapler Identification* section.

## Exposure Assessment: Surgical stapler identification

Exposure classification will be determined using the detailed billing records and hospital charge master descriptions available in the Premier Health Database. We will apply a prespecified text-string dictionary (brand, model, and common abbreviations) derived from official manufacturer product listings to identify stapler type. Public product resources are cited for transparency and to anchor the dictionary development: Intuitive SureForm staplers [2], Ethicon Echelon series (comparison chart with models/ordering codes) [19], and Medtronic staplers (portfolio pages with ordering information) [20]:

• Robotic Staplers, identified explicitly by product descriptors such as Intuitive Surgical's "SureForm" and text-based iterations and abbreviations.

• Bedside Staplers, further classified as:

  ◦ Powered staplers, including Medtronic's "Signia" and Ethicon's "Echelon Powered" staplers, commonly fired by bedside assistants in robotic surgery, and

  ◦ Manual staplers, including Medtronic's "GIA" and Ethicon's "Echelon" staplers.

The capture window is limited to the intraoperative period. Records with ambiguous or conflicting text will be excluded from the primary analysis. External thoracic and general surgeons will review the dictionary and a sample of classified records to confirm face validity.

For procedures in which both console-fired robotic staplers and bedside-operated staplers are used within the same operation ("mixed-use" cases), the encounter will be classified in the robotic stapler cohort. This approach reflects standard clinical workflow wherein robotic stapling is typically attempted first, with bedside stapling employed subsequently only as needed. Classifying mixed-use cases as robotic minimizes the risk of misclassification, particularly for situations in which bedside stapling is introduced after an initial robotic firing (e.g., during potential conversion events). To assess the robustness of this assumption, we will conduct a sensitivity analysis that excludes mixed-use encounters entirely and compares resulting estimates with those from the primary analysis.

All the texts strings prespecified to identify the above staplers are listed in the *Supporting information* (**S1 Table**) section.

## Outcomes

The primary outcome is postoperative leak: air leak for lung resection; anastomotic/staple-line leak for colorectal, gastrectomy, and bariatric procedures. Key secondary outcomes include complications, conversion to open surgery, blood

**Table 1. Codes used to identify robotic-assisted procedures.**

| Procedure | Code Type | Codes |
|---|---|---|
| Robotic procedure | CPT/HCPCS | S2900 |
| | ICD-10-PCS | 8E0W0CZ, 8E0W3CZ, 8E0W4CZ, 8E0W7CZ, 8E0W8CZ, 8E0WXCZ |
| **Lung resection** | | |
| Lobectomy | CPT/HCPCS | 32480, 32482, 32486, 32663, 32668, 32670 |
| | ICD-10-PCS | 0BTC0ZZ, 0BTC4ZZ, 0BTD0ZZ, 0BTD4ZZ, 0BTF0ZZ, 0BTF4ZZ, 0BTG0ZZ, 0BTG4ZZ, 0BTJ0ZZ, 0BTJ4ZZ |
| Segmentectomy and wedge resection | CPT/HCPCS | 32505, 32666, 32484, 32669 |
| | ICD-10-PCS | 0BBC0ZX, 0BBC0ZZ, 0BBC4ZX, 0BBC4ZZ, 0BBD0ZX, 0BBD0ZZ, 0BBD4ZX, 0BBD4ZZ, 0BBF0ZX, 0BBF0ZZ, 0BBF4ZX, 0BBF4ZZ, 0BBG0ZX, 0BBG0ZZ, 0BBG4ZX, 0BBG4ZZ, 0BBH0ZX, 0BBH0ZZ, 0BBH4ZX, 0BBH4ZZ, 0BBJ0ZX, 0BBJ0ZZ, 0BBJ4ZX, 0BBJ4ZZ, 0BBK0ZX, 0BBK0ZZ, 0BBK4ZX, 0BBK4ZZ, 0BBL0ZX, 0BBL0ZZ, 0BBL4ZX, 0BBL4ZZ, 0BBM0ZX, 0BBM0ZZ, 0BBM4ZX, 0BBM4ZZ |
| **Colon resection** | | |
| Hemicolectomy – Left | CPT/HCPCS | 44140, 44204 |
| | ICD-10-PCS | 0D1M0Z4, 0D1M4Z4, 0DBG0ZZ, 0DBG4ZZ, 0DBM0ZZ, 0DBM4ZZ, 0DTG0ZZ, 0DTG4ZZ, 0DTM0ZZ, 0DTM4ZZ |
| Hemicolectomy – Right | CPT/HCPCS | 44160, 44205 |
| | ICD-10-PCS | 0DBF0ZZ, 0DBF4ZZ, 0DBK0ZZ, 0DBK4ZZ, 0DTF0ZZ, 0DTF4ZZ, 0DTH0ZZ, 0DTH4ZZ, 0DTK0ZZ, 0DTK4ZZ |
| Sigmoid Colectomy | ICD-10-PCS | 0D1N0Z4, 0D1N0ZP, 0D1N4Z4, 0D1N4ZP, 0DBN0ZZ, 0DBN4ZZ, 0DTN0ZZ, 0DTN4ZZ |
| **Rectal resection** | | |
| Abdominoperi-neal Resection | CPT/HCPCS | 45110, 45395, 45397 |
| Low Anterior Resection | CPT/HCPCS | 44145, 44146, 44207, 44208, 45111, 45112, 45114, 45116, 45119, 45123 |
| | ICD-10-PCS | 0D1B0ZQ, 0D1B4ZQ, 0DBP0ZZ, 0DBP4ZZ, 0DBQ0ZZ, 0DBQ4ZZ, 0DTP0ZZ, 0DTP4ZZ, 0DTQ0ZZ, 0DTQ4ZZ |
| **Gastrectomy** | | |
| Total gastrectomy | CPT/HCPCS | 43620, 43621, 43622 |
| | ICD-10-PCS | 0DT60ZZ, 0DT64ZZ, 0DT70ZZ, 0DT74ZZ |
| Distal gastrectomy | CPT/HCPCS | 43631, 43632, 43633, 43634, 43635 |
| | ICD-10-PCS | 0DB70ZZ, 0DB74ZZ |
| Proximal gastrectomy | CPT/HCPCS | 43638, 43639 |
| | ICD-10-PCS | 0DB60ZZ, 0DB64ZZ |
| **Bariatric surgery** | | |
| Roux-en-Y gas-tric bypass | CPT/HCPCS | 43633, 43644, 43645, 43846, 43847 |
| | ICD-10-PCS | 0D16079, 0D1607A, 0D160J9, 0D160JA, 0D160K9, 0D160KA, 0D160Z9, 0D160ZA, 0D1607A, 0D160JA, 0D160KA, 0D160ZA, 0D1687A, 0D168JA, 0D168K9, 0D168KA, 0D168ZA, 0D168Z9, 0D16879, 0D1687A, 0D168ZA, 0D168J9, 0D16479, 0D1647A, 0D164J9, 0D164JA, 0D164K9, 0D164KA, 0D164Z9, 0D164ZA |
| Sleeve gastrectomy | CPT/HCPCS | 43775 |
| | ICD-10-PCS | 0DB64Z3, 0DQ60ZZ, 0DQ63ZZ, 0DQ67ZZ, 0DB60Z3 |
| Biliopancreatic diversion/ duo-denal switch | CPT/HCvPCS | 43845 |
| | ICD-10-PCS | 0D190Z9, 0DB60ZZ, 0DB80ZZ |

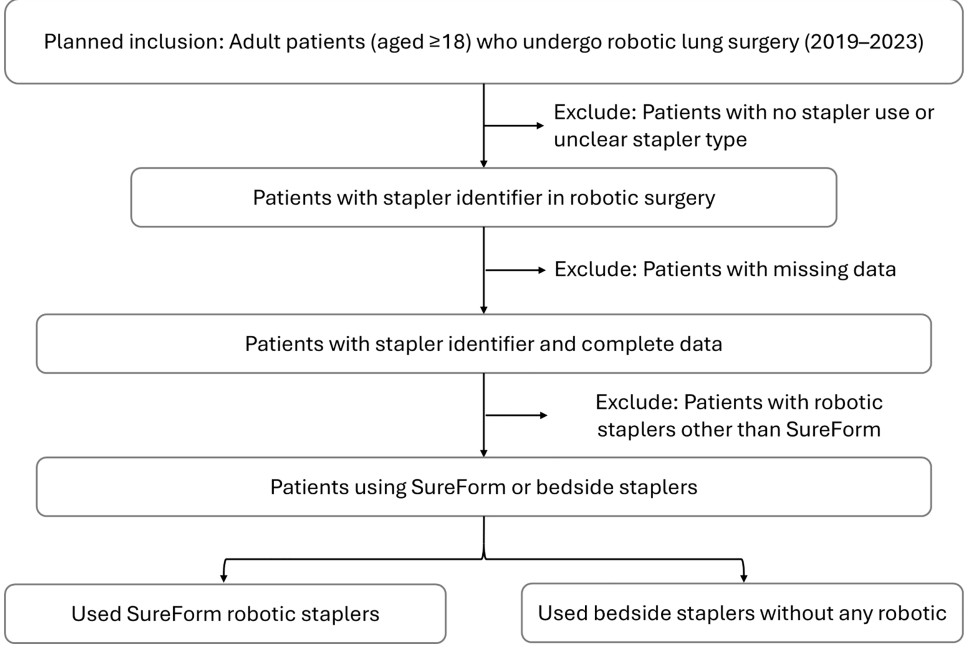

**Fig 1. Cohort Creation Flow Diagram.**

transfusion, operative time, hospital length of stay (LOS), and costs (index hospitalization and 30-day perioperative period). Operative time and costs will be obtained from billing/chargemaster data. LOS will be calculated as days from admission to discharge. Complications will vary by specialty. For lung surgery, complications include air leaks, bleeding, fistulas, pneumonia, acute post-hemorrhagic anemia, pneumothorax, and infections. For colorectal, gastrectomy, and bariatric procedures, complications include anastomotic or staple line leaks, peritoneal abscess, ileus, intestinal ischemia, peritonitis, urinary tract infections, surgical site infections, intra-abdominal bleeding, deep vein thrombosis, and bleeding.

The primary outcomes will be identified using the International Classification of Diseases, Tenth Revision, Clinical Modification (ICD-10-CM) (see **Table 2**). The incidence of outcomes will be defined as any occurrence of each specified event during hospitalization.

Secondary outcomes will focus on health care cost and health care resource utilization (HCRU) including total operative time, transfusion requirements, and hospital length of stay (LOS). Operative time will be calculated from the detailed billing records and hospital charge master descriptions available in the PHD, and LOS will be defined as the number of days between admission and discharge during hospitalization. Transfusions will be identified through the ICD-10-PCS and CPT codes in the PHD (**Table 2**).

Cost variables will include both index cost and 30-day perioperative cost from the hospital perspective. Index cost will summarize all the billing items during the index hospitalization, and 30-day perioperative cost will be calculated as the sum of the index cost and any costs within 30 days (pre and post) from the same hospital for each patient [21,22]. Stapler-related component costs will be reported separately from other cost categories.

## Baseline Characteristics

Baseline characteristics will be determined at the time of index hospitalization and will include demographic, clinical, institutional, and professional variables. Demographics will cover age, sex (male/female), race (white, black, other/unknown), and payor type (commercial, Medicare, Medicaid, and other). Clinical variables evaluating patients' baseline health status

**Table 2. Codes used to define outcomes.**

| Outcomes | Code Type | Code |
|---|---|---|
| Transfusion requirement | CPT/HCPCS | 36430 |
| | ICD-10-PCS | 30230M0, 30230M1, 30233H0, 30233H1, 30233K1, 30233L1, 30233M0, 30233M1, 30233N0, 30233N1, 30233P1, 30233R1, 30240H0, 30240H1, 30240K0, 30240K1, 30240L0, 30240L1, 30240M0, 30240M1, 30240N0, 30240N1, 30240P0, 30240P1, 30240R0, 30240R1, 30243H0, 30243H1, 30243K0, 30243K1, 30243L0, 30243L1, 30243M0, 30243M1, 30243N0, 30243N1, 30243P0, 30243P1, 30243R0, 30243R1, 30250H0, 30250H1, 30250K0, 30250K1, 30250L0, 30250L1, 30250M0, 30250M1, 30250N0, 30250N1, 30250P0, 30250P1, 30250R0, 30250R1, 30253H0, 30253H1, 30253K0, 30253K1, 30253L0, 30253L1, 30253M0, 30253M1, 30253N0, 30253N1, 30253P0, 30253P1, 30253R0, 30253R1, 30260H0, 30260H1, 30260K0, 30260K1, 30260L0, 30260L1, 30260M0, 30260M1, 30260N0, 30260N1, 30260P0, 30260P1, 30260R0, 30260R1, 30263H0, 30263H1, 30263K0, 30263K1, 30263L0, 30263L1, 30263M0, 30263M1, 30263N0, 30263N1, 30263P0, 30263P1, 30263R0, 30263R1 |
| Conversion to open surgery | ICD-10-CM | Z53.31, Z53.32, Z53.39 |
| **Complications for lung surgery** | | |
| Air leak | ICD-10-CM | J93.82, J95.812 |
| Bleeding | ICD-10-CM | D62, J95.61, J95.830, J95.860, J95.862, L76.02, M96.811, T88.8XXA |
| Fistula | ICD-10-CM | J86.0, T81.83XA, T81.83XD, T81.83XS |
| Pneumonia | ICD-10-CM | J13, J14, J15.0, J15.1, J15.211, J15.212, J15.4, J15.7, J15.8, J15.9, J16.8, J18.0, J18.1, J18.9, J69.0, J69.8, J95.89 |
| Acute post hemor-rhagic anemia | ICD-10-CM | D62 |
| Pneumothorax | ICD-10-CM | J93.83, J93.9, J95.811 |
| Infection | ICD-10-CM | A40.3, A40.9, A41.01, A41.02, A41.1, A41.2, A41.3, A41.4, A41.50, A41.51, A41.52, A41.53, A41.59, A41.89, A41.9, J67.7, J95.851, L03.313, L03.319, R65.20, R65.21, R78.81, T81.4XXA, T81.4XXD, T81.4XXS |
| **Complications for colorectal, gastric, and bariatric surgery** | | |
| Anastomotic leak | ICD-10-CM | K63.0, K63.2, K65.1, K91.89, Y83.2 |
| Peritoneal abscess | ICD-10-CM | K65.1 |
| Ileus | ICD-10-CM | K56.0, K56.1, K56.2, K56.41, K56.5, K56.50, K56.51, K56.52, K56.60, K56.600, K56.601, K56.609, K56.69, K56.690, K56.691, K56.699, K56.7, K91.3 |
| Intestinal ischemia | ICD-10-CM | K55.0, K55.019, K55.021, K55.022, K55.029, K55.031, K55.032, K55.039, K55.041, K55.049, K55.059, K55.069, K55.1, K55.20, K55.21, K55.8, K55.9 |
| Peritonitis | ICD-10-CM | K65.0, K65.1, K65.2, K65.3, K65.4, K65.8, K65.9 |
| Surgical site infection | ICD-10-CM | T81.4, T81.5, T81.6, T82.7, T85.7, T88.0 |
| Urinary tract infection | ICD-10-CM | N10, N30.00, N30.01, N30.10, N30.20, N30.30, N30.40, N30.41, N30.80, N30.81, N30.90, N30.91, N34.2, N39.0, N39.3, N39.41, N39.43, N39.45, N39.46, N39.490, N39.498 |
| Bleeding | ICD-10-CM | D62, R58, K92.2, K92.1, K91.840, K91.841, K91.870, K91.871, N99.62, N99.820, N99.821 |
| Intra-abdominal bleeding | ICD-10-CM | K66.1 |
| Deep vein thrombosis (DVT) | ICD-10-CM | I80.1-I80,9, I82.8, I82.9, O22.3, O22.8, O22.9, O87.1, O87.8, O87.9 |

will include obesity, Charlson Comorbidity Index (CCI) excluding cancer, malignancy (benign/malignant), and disease type. Institutional factors will include hospital bed size (<300, 300–399, 400–499, 500+) and provider vicinity (urban/rural). Professional variables will evaluate whether the surgery was performed by a surgical specialist or a surgical generalist. We will also account for learning-curve effects by including measures of provider experience: physician volume, defined as the total number of robotic-assisted procedures that each surgeon performs within the relevant specialty during the study period; and hospital volume, defined analogously as the total number of robotic-assisted procedures performed at each hospital within that specialty over the same interval.

## Statistical analysis

Propensity scores (PS) will be generated via multivariable logistic regression incorporating all identified covariates. Overlap weighting, an advanced PS weighting method that emphasizes individuals that are most comparable between exposure groups, will be applied to achieve optimal covariate balance, reducing potential selection bias [23]. Covariates considered for adjustment will include demographics, clinical, institutional, professional variables, and procedure types. After applying overlap weighting, standardized mean differences (SMD) will be assessed to confirm successful covariate balancing, with a target SMD of less than 0.10. If adequate balance cannot be achieved using OW, propensity score matching (PSM) will be performed as an alternative adjustment method. If PSM also fails to achieve sufficient balance, multivariable generalized linear mixed models (GLMMs) will be applied to adjust for confounding factors that differ between stapler groups and are associated with the outcomes.

Descriptive statistics will summarize baseline characteristics across stapler groups using means (standard deviation) or medians (interquartile range) for continuous variables, and frequencies (proportions) for categorical variables. Statistical comparisons of categorical outcomes between robotic and bedside stapler groups will be performed using Chi-squared test, or Fisher's exact test, and continuous outcomes will be analyzed using Welch's t-test or Mann-Whitney U test, depending on the data distribution.

Sensitivity analysis will be performed by excluding mixed-use cases, defined as encounters in which both robotic and bedside staplers were used in the same operation. Results from this sensitivity analysis will be compared with those from the main analysis to evaluate consistency of findings.

All statistical tests will be two-sided, with an alpha level of 0.05, and p-values $< .05$ will be considered statistically significant. Statistical analyses will be conducted utilizing R software (version 4.4.1; R Foundation for Statistical Computing, Vienna, Austria) [24].

## Ethical Considerations

This protocol covers a study using retrospective, fully de-identified patient data from the Premier Healthcare Database. Given the nature of the data, Institutional Review Board exemption under HIPAA privacy regulations is anticipated, with no requirement for individual patient informed consent.

## Limitations

This protocol has several anticipated limitations. First, the study will use data from the PHD, a large U.S. hospital-based database. As a result, findings may not be generalizable to other countries or health systems with different care delivery structures or coding practices. Second, because the analysis will rely on administrative and billing data, certain clinical variables such as tumor stage, cartridge color or staple height selection, waiting time before firing, and other intraoperative technical factors, or surgeon decision-making processes may not be fully captured. Third, spirometry data such as $FEV_1$ and DLCO are not available in PHD. These measures are important for assessing baseline pulmonary function and could influence the risk of postoperative air leak. We will therefore interpret air-leak findings cautiously and explicitly acknowledge this limitation. Pneumothorax will also be examined as a related postoperative pulmonary outcome. Finally, during the protocol design stage, all authors are employees of Intuitive Surgical, the manufacturer of the robotic system evaluated in this research. To minimize potential bias, all analyses will follow this pre-specified statistical analysis plan, regardless of whether results favor the sponsor. Additionally, the analytic methods and exposure definitions will undergo external clinician review by independent thoracic and general surgeons not affiliated with the company to ensure objectivity and transparency.

## Dissemination Plan

Results will be disseminated through submission to peer-reviewed journals and presentations at relevant national and international meetings and conferences.

## Supporting information

**S1 Table. Strings for Identification Staplers.**
(DOCX)

## Author contributions

**Conceptualization:** Yu-Ting Chi, Peng-Lin Lin.

**Data curation:** Yu-Ting Chi.

**Methodology:** Yu-Ting Chi, Peng-Lin Lin.

**Project administration:** Peng-Lin Lin.

**Validation:** Peng-Lin Lin.

**Visualization:** Yu-Ting Chi.

**Writing – original draft:** Yu-Ting Chi, Peng-Lin Lin.

**Writing – review & editing:** Naomi C Hamm, Shih-Hao Lee, Minkyung Shin, Yuki Liu, I-Fan Shih, Feibi Zheng, Ben Forrest, Peng-Lin Lin.

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
