## [Decision Letter · Decision Letter 0]

20 Oct 2025

Dear Dr. Lin,

Thank you for submitting your manuscript to PLOS ONE. After careful consideration, we feel that it has merit but does not fully meet PLOS ONE’s publication criteria as it currently stands. Therefore, we invite you to submit a revised version of the manuscript that addresses the points raised during the review process.

We look forward to receiving your revised manuscript.

Kind regards,

Suphakarn Techapongsatorn, M.D., Ph.D.

Academic Editor

PLOS ONE

Journal Requirements:

2. Thank you for stating the following in the Competing Interests/Financial Disclosure section:

I have read the journal's policy and the authors of this manuscript have the following competing interests:

All authors are employees of Intuitive Surgical.

Yu-Ting Chi, Naomi C Hamm PhD, Shih-Hao Lee, Minkyung Shin, Yuki Liu, I-Fan Shih, Feibi Zheng, Ben Forrest, and Peng-Lin Lin receive salaries from Intuitive Surgical.

Additionally, Shih-Hao Lee, Minkyung Shin, Yuki Liu, I-Fan Shih, Feibi Zheng, Ben Forrest, and Peng-Lin Lin hold restricted stock units (RSUs) from Intuitive Surgical.

We note that one or more of the authors are employed by a commercial company: Intuitive Surgical

5. Please amend your list of authors on the manuscript to ensure that each author is linked to an affiliation. Authors’ affiliations should reflect the institution where the work was done (if authors moved subsequently, you can also list the new affiliation stating “current affiliation:….” as necessary).

Additional Editor Comments:

After careful evaluation of your manuscript, the reviewers have provided constructive comments and suggestions for improvement.

Please revise your manuscript accordingly. We hope that these comments will assist you in enhancing the quality and scientific rigour of your work, ensuring that it aligns with the journal’s requirements.

Reviewers' comments:

Reviewer's Responses to Questions

**Comments to the Author**

1. Does the manuscript provide a valid rationale for the proposed study, with clearly identified and justified research questions?

Reviewer #1: Partly

Reviewer #2: Yes

Reviewer #3: Yes

Reviewer #4: Yes

2. Is the protocol technically sound and planned in a manner that will lead to a meaningful outcome and allow testing the stated hypotheses?

Reviewer #1: Yes

Reviewer #2: Yes

Reviewer #3: Yes

Reviewer #4: Yes

3. Is the methodology feasible and described in sufficient detail to allow the work to be replicable?

Reviewer #1: Yes

Reviewer #2: Yes

Reviewer #3: Yes

Reviewer #4: Yes

4. Have the authors described where all data underlying the findings will be made available when the study is complete?

Reviewer #1: Yes

Reviewer #2: Yes

Reviewer #3: Yes

Reviewer #4: No

5. Is the manuscript presented in an intelligible fashion and written in standard English?

Reviewer #1: No

Reviewer #2: Yes

Reviewer #3: Yes

Reviewer #4: Yes

You may also provide optional suggestions and comments to authors that they might find helpful in planning their study.

Reviewer #1: the abstract, introduction and methods needs to be rewritten as a completed study...currently reads as a proposal.

To strengthen the study, I'd recommend adding an independent oversight mechanism to minimize sponsor bias concerns. The analysis would benefit from sensitivity analyses that exclude mixed-use stapler cases. I'd also suggest narrowing the primary outcomes to focus on those genuinely impacted by stapler type.

The methodology section needs more detail on misclassification risk and how you validated your approach. For the economic analysis, please separate stapler acquisition and maintenance costs from total costs - this distinction matters.

Finally, the discussion needs expansion, particularly around study limitations. Be upfront about generalizability issues and address the elephant in the room - industry involvement and how it might affect interpretation.

These changes would transform a good paper into one that's truly robust and defensible.

Reviewer #2: Good Protocol. Just a few comments:

1. Is there a clinician in the investigators?

2. Statistical Analysis will require more extensive detailing, given the huge number of cases and variables. Perhaps a Statician should be included in the team

3. Images, patents?, model numbers, FDA approvals, etc of the device should be included

Reviewer #3: Congratulations to the authors for their work and for their future work. Evaluating the costs of healthcare procedures is inseparable from the results obtained, and especially today, we cannot afford to waste money. The focus of the work is very interesting, and the method used in statistically evaluating current research is excellent. I believe it is appropriate to use different divisions and approaches to certify results in surgical specialties.

The methodology and writing are appropriate, as are the references.

Please make a minor revision to the descriptive specifications of the analyzed works:

Specify whether the surgeon's experience, learning curve, or different staplers influenced the evaluated works (points for consideration regarding correct use: waiting times before suturing; types of staples with different colors depending on the tissue).

Good luck, and thank you.

Reviewer #4: Dear Authors,

thanks a lot for submitting this protocol for publication in PLOS One. The study is very promising and interesting, especially the inclusion of the costs as outcomes. I have only one comment regarding the lung resection patients: it's mandatory that you collect the FEV1 and DLCO before surgery otherwise an analysis of the air leak as complications doesn't make any sense (if you have only patients with bad lung function in one group and patients with perfect lung function in the other group would be a big bias).

Thanks and good luck!

**Do you want your identity to be public for this peer review?** For information about this choice, including consent withdrawal, please see our Privacy Policy

Reviewer #1: No

Reviewer #2: **Yes: ** Mahmoud Elfiky

Reviewer #3: No

Reviewer #4: No

---

## [Author Response · Author response to Decision Letter 1]

13 Nov 2025

Reviewer #1:

1. the abstract, introduction and methods needs to be rewritten as a completed study...currently reads as a proposal.

Response:

This submission is a study protocol. To prevent confusion, we now explicitly state protocol status in the Abstract and Introduction and keep the Methods in future tense while providing full operational detail.

2. To strengthen the study, I'd recommend adding an independent oversight mechanism to minimize sponsor bias concerns.

Response:

For each procedure and related publication, we will work with independent external surgeons who will review and approve the final analysis plan before data lock and provide independent input on the results. Their involvement helps ensure scientific rigor and reduce potential bias.

All analyses will follow the pre-specified statistical analysis plan in this protocol, regardless of whether the findings favor the sponsor. This approach is designed to support transparency and protect the scientific integrity of the study.

3. The analysis would benefit from sensitivity analyses that exclude mixed-use stapler cases.

Response:

Thank you for the helpful suggestion. In our primary analysis, we classify mixed-use cases—operations in which both console-fired robotic staplers and bedside-operated staplers are used—into the robotic stapler group. This reflects typical surgical practice, where robotic stapling is attempted first and bedside stapling is added only if needed (including in situations related to potential conversion). Removing these cases from the robotic cohort may incorrectly exclude scenarios where a conversion occurred after a robotic firing.

To address this concern, we have included a sensitivity analysis that excludes all mixed-use cases and compares results with the primary analysis.

We have added the following clarification to the Exposure Assessment: Surgical Stapler Identification section of the Methods:

“For procedures in which both console-fired robotic staplers and bedside-operated staplers are used within the same operation (“mixed-use” cases), the encounter will be classified in the robotic stapler cohort. This approach reflects standard clinical workflow wherein robotic stapling is typically attempted first, with bedside stapling employed subsequently only as needed. Classifying mixed-use cases as robotic minimizes the risk of misclassification, particularly for situations in which bedside stapling is introduced after an initial robotic firing (e.g., during potential conversion events). To assess the robustness of this assumption, we will conduct a sensitivity analysis that excludes mixed-use encounters entirely and compares resulting estimates with those from the primary analysis.”

We also have added the following paragraph to the Statistical Analysis section:

“Sensitivity analysis will be performed by excluding mixed-use cases, defined as encounters in which both robotic and bedside staplers were used in the same operation. Results from this sensitivity analysis will be compared with those from the main analysis to evaluate consistency of findings.”

4. I'd also suggest narrowing the primary outcomes to focus on those genuinely impacted by stapler type.

Responses:

As you advised, we narrowed the primary outcome to postoperative leak (air leak for lung resection; anastomotic leak for colorectal, gastrectomy, and bariatric). Conversion, transfusion, operative time, LOS, and costs are now key secondary outcomes. Details are in Methods—Exposure Assessment, Outcomes, and Statistical Analysis.

5. The methodology section needs more detail on misclassification risk and how you validated your approach.

Response:

Thank you for the helpful comment. We have clarified how stapler types are identified and how potential misclassification will be minimized and validated.

Stapler identification will rely on a prespecified dictionary derived from official manufacturer product listings, including brand names, model identifiers, and ordering codes. Publicly available product references from Intuitive Surgical, Ethicon (Johnson & Johnson), and Medtronic are cited in the Supplemental Materials for transparency:

• Intuitive SureForm Datasheet (https://www.intuitive.com/en-gb/-/media/ISI/Intuitive/Pdf/stapling-sureform-data-sheet-europe-1061915.pdf?utm_source=chatgpt.com)

• Ethicon Laparoscopic Stapler Comparison Chart (https://www.jnjmedtech.com/system/files/pdf/laparoscopic-stapler-comparison-chart.pdf)

• Medtronic Stapler Ordering Information (GIA, Signia, and related models) (https://www.medtronic.com/en-us/healthcare-professionals/products/surgical-stapling/surgical-staplers.html)

Ambiguous or conflicting entries will be excluded from the analysis. In addition, external thoracic and general surgeons will review the dictionary and a sample of classified records to confirm face validity and ensure accurate categorization.

6. For the economic analysis, please separate stapler acquisition and maintenance costs from total costs - this distinction matters.

Responses:

Thank you for the suggestion. Where available in the charge-level data, stapler component costs (e.g., device and reload charges) will be reported separately from other cost categories. However, capital and maintenance costs are not captured at the encounter level in the database and therefore cannot be separated; we will note this as a limitation in the formal analyses.

We have added this sentence to the Outcome section:

Stapler-related component costs will be reported separately from other cost categories.

7. Finally, the discussion needs expansion, particularly around study limitations. Be upfront about generalizability issues and address the elephant in the room - industry involvement and how it might affect interpretation.

These changes would transform a good paper into one that's truly robust and defensible.

Responses:

Thank you for this important comment. As this is a study protocol, no results are yet available, but we agree that key limitations and transparency regarding funding should be clearly described.

We have added a limitation section in the protocol:

This protocol has several anticipated limitations. First, the study will use data from the PHD, a large U.S. hospital-based database. As a result, findings may not be generalizable to other countries or health systems with different care delivery structures or coding practices. Second, because the analysis will rely on administrative and billing data, certain clinical variables such as tumor stage, intraoperative technical factors, or surgeon decision-making processes may not be fully captured. Third, during the protocol design stage, all authors are employees of Intuitive Surgical, the manufacturer of the robotic system evaluated in this research. To minimize potential bias, all analyses will follow this pre-specified statistical analysis plan, regardless of whether results favor the sponsor. Additionally, the analytic methods and exposure definitions will undergo external clinician review by independent thoracic and colorectal surgeons not affiliated with the company to ensure objectivity and transparency.

Reviewer #2:

1. Is there a clinician in the investigators?

Responses:

Yes, for the formal analysis and the publications for each procedure, we will collaborate with external surgeons to review the final analysis plan prior to data lock and to adjudicate results.

2. Statistical Analysis will require more extensive detailing, given the huge number of cases and variables. Perhaps a Statician should be included in the team.

Responses:

P.-L. Lin, S.-H. Lee, Y. Liu, and Y.-T. Chi are senior biostatisticians with extensive experience in large-scale database analyses and observational comparative studies. We have expanded the Statistical Analysis section to provide more details. Specifically, we now describe (i) the inclusion of a sensitivity analysis, (ii) the use of propensity score matching (PSM) if overlap weighting (OW) does not achieve adequate cohort balance (standardized mean difference [SMD] < 0.10), and (iii) the application of generalized linear mixed models (GLMM) if balance remains insufficient after PSM. We also elaborated on statistical testing methods for both categorical and continuous variables.

3. Images, patents?, model numbers, FDA approvals, etc of the device should be included

Responses:

Thank you for the suggestion. All staplers included in this protocol are commercially available and FDA-approved. To maintain readability and avoid promotional appearance, we provide links to official product pages that list detailed model numbers and ordering information instead of reproducing all device details here.

Robotic staplers correspond to Intuitive SureForm™ staplers; bedside staplers include Ethicon Echelon™ (Johnson & Johnson) and Medtronic Signia™/GIA™ staplers.

Official device information and ordering references are cited in the Supplemental Materials and include:

• Intuitive SureForm Datasheet (https://www.intuitive.com/en-gb/-/media/ISI/Intuitive/Pdf/stapling-sureform-data-sheet-europe-1061915.pdf?utm_source=chatgpt.com)

• Ethicon Laparoscopic Stapler Comparison Chart (https://www.jnjmedtech.com/system/files/pdf/laparoscopic-stapler-comparison-chart.pdf)

• Medtronic Stapler Ordering Information (GIA, Signia, and related models) (https://www.medtronic.com/en-us/healthcare-professionals/products/surgical-stapling/surgical-staplers.html)

Reviewer #3:

Congratulations to the authors for their work and for their future work. Evaluating the costs of healthcare procedures is inseparable from the results obtained, and especially today, we cannot afford to waste money. The focus of the work is very interesting, and the method used in statistically evaluating current research is excellent. I believe it is appropriate to use different divisions and approaches to certify results in surgical specialties.

The methodology and writing are appropriate, as are the references.

Please make a minor revision to the descriptive specifications of the analyzed works:

Specify whether the surgeon's experience, learning curve, or different staplers influenced the evaluated works (points for consideration regarding correct use: waiting times before suturing; types of staples with different colors depending on the tissue).

Good luck, and thank you.

Responses:

Thank you for the thoughtful suggestion. We have clarified in the Statistical Analysis section that surgeon and hospital volumes will be included as proxies for the learning curve and surgical experience. Currently, it remains challenging (or simply not possible) to identify intraoperative details such as cartridge color selection, waiting time before firing, or staple type based on tissue characteristics in the Premier Healthcare Database (PHD). These limitations have been acknowledged in the Limitations section.

Reviewer #4:

Dear Authors,

thanks a lot for submitting this protocol for publication in PLOS One. The study is very promising and interesting, especially the inclusion of the costs as outcomes. I have only one comment regarding the lung resection patients: it's mandatory that you collect the FEV1 and DLCO before surgery otherwise an analysis of the air leak as complications doesn't make any sense (if you have only patients with bad lung function in one group and patients with perfect lung function in the other group would be a big bias).

Thanks and good luck!

Responses:

Thank you very much for this valuable comment. Spirometry data, including FEV1 and DLCO, are not available in the Premier Healthcare Database (PHD). To partially address this limitation, we included pneumothorax in the complication list as a related indicator of postoperative pulmonary status. We will also explicitly acknowledge the absence of preoperative lung function data as a limitation and interpret air-leak findings with caution, recognizing the potential confounding effect of baseline pulmonary function differences between groups.

---

## [Editor Report · Decision Letter 1]

16 Nov 2025

Dear Dr. Lin,

Thank you for submitting your manuscript to PLOS ONE. After careful consideration, we feel that it has merit but does not fully meet PLOS ONE’s publication criteria as it currently stands. Therefore, we invite you to submit a revised version of the manuscript that addresses the points raised during the review process.

To facilitate my review of the changes made in response to the reviewers’ comments, may I kindly request that you highlight or track the specific modifications in the revised manuscript compared with the original version?

This will help ensure that all revisions have been appropriately addressed and will make it much easier for me to verify the updated sections.

We look forward to receiving your revised manuscript.

Kind regards,

Suphakarn Techapongsatorn, M.D., Ph.D.

Academic Editor

PLOS ONE
---

## [Author Response · Author response to Decision Letter 2]

20 Nov 2025

Reviewer #1:

1. the abstract, introduction and methods needs to be rewritten as a completed study...currently reads as a proposal.

Response:

This submission is a study protocol. To prevent confusion, we now explicitly state protocol status in the Abstract and Introduction and keep the Methods in future tense while providing full operational detail.

2. To strengthen the study, I'd recommend adding an independent oversight mechanism to minimize sponsor bias concerns.

Response:

For each procedure and related publication, we will work with independent external surgeons who will review and approve the final analysis plan before data lock and provide independent input on the results. Their involvement helps ensure scientific rigor and reduce potential bias.

All analyses will follow the pre-specified statistical analysis plan in this protocol, regardless of whether the findings favor the sponsor. This approach is designed to support transparency and protect the scientific integrity of the study.

3. The analysis would benefit from sensitivity analyses that exclude mixed-use stapler cases.

Response:

Thank you for the helpful suggestion. In our primary analysis, we classify mixed-use cases—operations in which both console-fired robotic staplers and bedside-operated staplers are used—into the robotic stapler group. This reflects typical surgical practice, where robotic stapling is attempted first and bedside stapling is added only if needed (including in situations related to potential conversion). Removing these cases from the robotic cohort may incorrectly exclude scenarios where a conversion occurred after a robotic firing.

To address this concern, we have included a sensitivity analysis that excludes all mixed-use cases and compares results with the primary analysis.

We have added the following clarification to the Exposure Assessment: Surgical Stapler Identification section of the Methods:

“For procedures in which both console-fired robotic staplers and bedside-operated staplers are used within the same operation (“mixed-use” cases), the encounter will be classified in the robotic stapler cohort. This approach reflects standard clinical workflow wherein robotic stapling is typically attempted first, with bedside stapling employed subsequently only as needed. Classifying mixed-use cases as robotic minimizes the risk of misclassification, particularly for situations in which bedside stapling is introduced after an initial robotic firing (e.g., during potential conversion events). To assess the robustness of this assumption, we will conduct a sensitivity analysis that excludes mixed-use encounters entirely and compares resulting estimates with those from the primary analysis.”

We also have added the following paragraph to the Statistical Analysis section:

“Sensitivity analysis will be performed by excluding mixed-use cases, defined as encounters in which both robotic and bedside staplers were used in the same operation. Results from this sensitivity analysis will be compared with those from the main analysis to evaluate consistency of findings.”

4. I'd also suggest narrowing the primary outcomes to focus on those genuinely impacted by stapler type.

Responses:

As you advised, we narrowed the primary outcome to postoperative leak (air leak for lung resection; anastomotic leak for colorectal, gastrectomy, and bariatric). Conversion, transfusion, operative time, LOS, and costs are now key secondary outcomes. Details are in Methods—Exposure Assessment, Outcomes, and Statistical Analysis.

5. The methodology section needs more detail on misclassification risk and how you validated your approach.

Response:

Thank you for the helpful comment. We have clarified how stapler types are identified and how potential misclassification will be minimized and validated.

Stapler identification will rely on a prespecified dictionary derived from official manufacturer product listings, including brand names, model identifiers, and ordering codes. Publicly available product references from Intuitive Surgical, Ethicon (Johnson & Johnson), and Medtronic are cited in the Supplemental Materials for transparency:

• Intuitive SureForm Datasheet (https://www.intuitive.com/en-gb/-/media/ISI/Intuitive/Pdf/stapling-sureform-data-sheet-europe-1061915.pdf?utm_source=chatgpt.com)

• Ethicon Laparoscopic Stapler Comparison Chart (https://www.jnjmedtech.com/system/files/pdf/laparoscopic-stapler-comparison-chart.pdf)

• Medtronic Stapler Ordering Information (GIA, Signia, and related models) (https://www.medtronic.com/en-us/healthcare-professionals/products/surgical-stapling/surgical-staplers.html)

Ambiguous or conflicting entries will be excluded from the analysis. In addition, external thoracic and general surgeons will review the dictionary and a sample of classified records to confirm face validity and ensure accurate categorization.

6. For the economic analysis, please separate stapler acquisition and maintenance costs from total costs - this distinction matters.

Responses:

Thank you for the suggestion. Where available in the charge-level data, stapler component costs (e.g., device and reload charges) will be reported separately from other cost categories. However, capital and maintenance costs are not captured at the encounter level in the database and therefore cannot be separated; we will note this as a limitation in the formal analyses.

We have added this sentence to the Outcome section:

Stapler-related component costs will be reported separately from other cost categories.

7. Finally, the discussion needs expansion, particularly around study limitations. Be upfront about generalizability issues and address the elephant in the room - industry involvement and how it might affect interpretation.

These changes would transform a good paper into one that's truly robust and defensible.

Responses:

Thank you for this important comment. As this is a study protocol, no results are yet available, but we agree that key limitations and transparency regarding funding should be clearly described.

We have added a limitation section in the protocol:

This protocol has several anticipated limitations. First, the study will use data from the PHD, a large U.S. hospital-based database. As a result, findings may not be generalizable to other countries or health systems with different care delivery structures or coding practices. Second, because the analysis will rely on administrative and billing data, certain clinical variables such as tumor stage, intraoperative technical factors, or surgeon decision-making processes may not be fully captured. Third, during the protocol design stage, all authors are employees of Intuitive Surgical, the manufacturer of the robotic system evaluated in this research. To minimize potential bias, all analyses will follow this pre-specified statistical analysis plan, regardless of whether results favor the sponsor. Additionally, the analytic methods and exposure definitions will undergo external clinician review by independent thoracic and colorectal surgeons not affiliated with the company to ensure objectivity and transparency.

Reviewer #2:

1. Is there a clinician in the investigators?

Responses:

Yes, for the formal analysis and the publications for each procedure, we will collaborate with external surgeons to review the final analysis plan prior to data lock and to adjudicate results.

2. Statistical Analysis will require more extensive detailing, given the huge number of cases and variables. Perhaps a Statician should be included in the team.

Responses:

P.-L. Lin, S.-H. Lee, Y. Liu, and Y.-T. Chi are senior biostatisticians with extensive experience in large-scale database analyses and observational comparative studies. We have expanded the Statistical Analysis section to provide more details. Specifically, we now describe (i) the inclusion of a sensitivity analysis, (ii) the use of propensity score matching (PSM) if overlap weighting (OW) does not achieve adequate cohort balance (standardized mean difference [SMD] < 0.10), and (iii) the application of generalized linear mixed models (GLMM) if balance remains insufficient after PSM. We also elaborated on statistical testing methods for both categorical and continuous variables.

3. Images, patents?, model numbers, FDA approvals, etc of the device should be included

Responses:

Thank you for the suggestion. All staplers included in this protocol are commercially available and FDA-approved. To maintain readability and avoid promotional appearance, we provide links to official product pages that list detailed model numbers and ordering information instead of reproducing all device details here.

Robotic staplers correspond to Intuitive SureForm™ staplers; bedside staplers include Ethicon Echelon™ (Johnson & Johnson) and Medtronic Signia™/GIA™ staplers.

Official device information and ordering references are cited in the Supplemental Materials and include:

• Intuitive SureForm Datasheet (https://www.intuitive.com/en-gb/-/media/ISI/Intuitive/Pdf/stapling-sureform-data-sheet-europe-1061915.pdf?utm_source=chatgpt.com)

• Ethicon Laparoscopic Stapler Comparison Chart (https://www.jnjmedtech.com/system/files/pdf/laparoscopic-stapler-comparison-chart.pdf)

• Medtronic Stapler Ordering Information (GIA, Signia, and related models) (https://www.medtronic.com/en-us/healthcare-professionals/products/surgical-stapling/surgical-staplers.html)

Reviewer #3:

Congratulations to the authors for their work and for their future work. Evaluating the costs of healthcare procedures is inseparable from the results obtained, and especially today, we cannot afford to waste money. The focus of the work is very interesting, and the method used in statistically evaluating current research is excellent. I believe it is appropriate to use different divisions and approaches to certify results in surgical specialties.

The methodology and writing are appropriate, as are the references.

Please make a minor revision to the descriptive specifications of the analyzed works:

Specify whether the surgeon's experience, learning curve, or different staplers influenced the evaluated works (points for consideration regarding correct use: waiting times before suturing; types of staples with different colors depending on the tissue).

Good luck, and thank you.

Responses:

Thank you for the thoughtful suggestion. We have clarified in the Statistical Analysis section that surgeon and hospital volumes will be included as proxies for the learning curve and surgical experience. Currently, it remains challenging (or simply not possible) to identify intraoperative details such as cartridge color selection, waiting time before firing, or staple type based on tissue characteristics in the Premier Healthcare Database (PHD). These limitations have been acknowledged in the Limitations section.

Reviewer #4:

Dear Authors,

thanks a lot for submitting this protocol for publication in PLOS One. The study is very promising and interesting, especially the inclusion of the costs as outcomes. I have only one comment regarding the lung resection patients: it's mandatory that you collect the FEV1 and DLCO before surgery otherwise an analysis of the air leak as complications doesn't make any sense (if you have only patients with bad lung function in one group and patients with perfect lung function in the other group would be a big bias).

Thanks and good luck!

Responses:

Thank you very much for this valuable comment. Spirometry data, including FEV1 and DLCO, are not available in the Premier Healthcare Database (PHD). To partially address this limitation, we included pneumothorax in the complication list as a related indicator of postoperative pulmonary status. We will also explicitly acknowledge the absence of preoperative lung function data as a limitation and interpret air-leak findings with caution, recognizing the potential confounding effect of baseline pulmonary function differences between groups.

---

## [Decision Letter · Decision Letter 2]

3 Dec 2025

A study protocol for a multi-specialty observational cohort comparing robotic stapler and bedside stapler outcomes in robotic-assisted surgeries

PONE-D-25-28738R2

Dear Dr. Lin,

We’re pleased to inform you that your manuscript has been judged scientifically suitable for publication and will be formally accepted for publication once it meets all outstanding technical requirements.

Kind regards,

Suphakarn Techapongsatorn, M.D., Ph.D.

Academic Editor

PLOS ONE

Additional Editor Comments (optional):

Reviewers' comments:

Reviewer's Responses to Questions

**Comments to the Author**

1. Does the manuscript provide a valid rationale for the proposed study, with clearly identified and justified research questions?

Reviewer #2: Yes

Reviewer #4: Yes

2. Is the protocol technically sound and planned in a manner that will lead to a meaningful outcome and allow testing the stated hypotheses?

Reviewer #2: Yes

Reviewer #4: Yes

3. Is the methodology feasible and described in sufficient detail to allow the work to be replicable?

Reviewer #2: Yes

Reviewer #4: Yes

4. Have the authors described where all data underlying the findings will be made available when the study is complete?

Reviewer #2: Yes

Reviewer #4: Yes

5. Is the manuscript presented in an intelligible fashion and written in standard English?

Reviewer #2: Yes

Reviewer #4: Yes

You may also provide optional suggestions and comments to authors that they might find helpful in planning their study.

Reviewer #2: Thank you for addressing the concerns of al reviewers. This shows a commitment and integrity to present the work

Reviewer #4: dear authors, thanks for submitting a revised paper providing some changes about the limitations of the study protocol.

**Do you want your identity to be public for this peer review?** For information about this choice, including consent withdrawal, please see our Privacy Policy

Reviewer #2: **Yes: ** Mahmoud Elfiky

Reviewer #4: No

---

## [Editor Report · Acceptance letter]

PONE-D-25-28738R2

PLOS One

Dear Dr. Lin,

I'm pleased to inform you that your manuscript has been deemed suitable for publication in PLOS One. Congratulations! Your manuscript is now being handed over to our production team.

Kind regards,

on behalf of

Dr. Suphakarn Techapongsatorn

Academic Editor

PLOS One